# RELATIONAL CURRICULUM LEARNING FOR GRAPH NEURAL NETWORK

## ABSTRACT

Graph neural networks have achieved great success in representing structured data and its downstream tasks such as node classification. The key idea is to recursively propagate and aggregate information along the edges of a given graph topology. However, edges in real-world graphs often have varying degrees of difficulty, and some edges may even be noisy to the downstream tasks. Therefore, existing graph neural network models may lead to suboptimal learned representations because they usually consider every edge in a given graph topology equally. On the other hand, curriculum learning, which mimics the human learning principle of learning data samples in a meaningful order, has been shown to be effective in improving the generalization ability and robustness of representation learners by gradually proceeding from easy to more difficult samples during training. Unfortunately, most existing curriculum learning strategies are designed for independent data samples and cannot be trivially generalized to handle data with dependencies. In order to address these issues, in this paper we propose a novel curriculum learning method for structured data to leverage the various underlying difficulties of data dependencies to improve the quality of learned representations on structured data. Specifically, we design a learning strategy that gradually incorporates edges in a given graph topology into training according to their difficulty from easy to hard, where the degree of difficulty is measured by a self-supervised learning paradigm. We demonstrate the strength of our proposed method in improving the generalization ability and robustness of learned representations through extensive experiments on nine synthetic datasets and nine real-world datasets with different commonly used graph neural network models as backbone models.

## 1 INTRODUCTION

Learning powerful representations of data samples with dependencies has become a core paradigm for understanding the underlying network mechanisms and performing a variety of downstream tasks such as social network analysis (Wasserman et al., 1994) and recommendation systems (Ying et al., 2018; Fan et al., 2019). As a class of state-of-the-art representation learning methods, graph neural networks (GNNs) have received increasing attention in recent years due to their powerful ability to jointly model data samples and their dependencies in an end-to-end fashion. Typically, GNNs treat data samples as nodes and their dependencies as edges, and then follow a neighborhood aggregation scheme to learn data sample representations by recursively transforming and aggregating information from their neighboring samples.

On the other hand, inspired by cognitive science studies (Elman, 1993; Rohde & Plaut, 1999) that humans can benefit from the sequence of learning basic (easy) concepts first and advanced (hard) concepts later, curriculum learning (CL) (Bengio et al., 2009; Kumar et al., 2010) suggests training a machine learning model with easy data samples first and then gradually introducing more hard samples into the model according to a designed pace, where the difficulty of samples can usually be measured by their training loss. Many previous studies have shown that this easy-to-hard learning strategy can effectively improve the generalization ability of the model (Bengio et al., 2009; Jiang et al., 2018; Han et al., 2018; Gong et al., 2016; Shrivastava et al., 2016; Weinshall et al., 2018). Furthermore, previous studies (Jiang et al., 2018; Han et al., 2018; Gong et al., 2016) have shown that CL strategies can increase the robustness of the model against noisy training samples. An intuitive

explanation is that in CL settings noisy data samples correspond to harder samples and CL learner spends less time with the harder (noisy) samples to achieve better generalization performance.

However, existing CL algorithms are typically designed for independent data samples (e.g. image) while designing effective CL strategy for data samples with dependencies has been largely under-explored. In dependent data, not only the difficulty of learning individual data samples can vary but also the difficulty of perceiving the dependencies between data samples. For example, in social networks, the connections between users with certain similar characteristics, such as geological location or interests, are usually considered as easy edges because their formation mechanisms are well expected. Some connections from unrelated users, such as advertisers or even bots, can usually be considered hard as they are not well expected or even noisy such that the likelihood of these connections can positively contribute to the downstream task, e.g. community detection, is relatively low. Therefore, as the previous CL strategies indicated that an easy-to-hard learning sequence on data samples can improve the learning performance, an intuitive question is whether a similar strategy on data dependencies that iteratively involving easy-to-hard edges in learning can also benefit.

Unfortunately, there exists no trivial way to directly generalize existing CL strategies on independent data to handle data dependencies due to several unique challenges: (1) **Difficulty in designing a feasible principle to select edges by properly quantifying how well they are expected**. Existing CL studies on independent data often use supervised computable metrics (e.g. training loss) to quantify sample complexity, but how to quantify how well the dependencies between data samples are expected which has no supervision is challenging. (2) **Difficulty in designing an appropriate pace to gradually involve edges based on model status**. Similar to the human learning process, the model should ideally retain a certain degree of freedom to adjust the curriculum according to its own learning status. It is extremely hard to design a general pacing policy suitable for different real-world scenarios. (3) **Difficulty in ensuring convergence and a numerical steady process for optimizing the model**. Since GNN models work by propagating and aggregating message information over the edges, due to discrete changes in the number of propagating messages by CL strategy, the learning process of incremental edges increases the difficulty of finding optimal parameters.

In order to address the aforementioned challenges, in this paper, we propose a novel CL algorithm named **R**elational **C**urriculum **L**earning (**RCL**) to improve the generalization ability of representation learners on data with dependencies. To address the first challenge, we propose a self-supervised learning approach to select the most $K$ easiest edges that are well expected by the model. Specifically, we jointly learn the node-level prediction task and estimate how well the edges are expected by measuring the relation between learned node embeddings. Second, to design an appropriate learning pace for gradually involving more edges in training, we present the learning process as a concise optimization model under the self-paced learning framework (Kumar et al., 2010), which lets the model gradually increase the number $K$ of selected edges according to its own status. Third, to ensure convergence of optimizing the model, we propose a proximal optimization algorithm with a theoretical convergence guarantee and an edge reweighting scheme to smooth the structure transition. Finally, we demonstrate the superior performance of RCL compared to state-of-the-art comparison methods through extensive experiments on both synthetic and real-world datasets.

## 2 RELATED WORK

**Curriculum learning (CL).**  Bengio et al. (2009) first proposed the idea of CL in the context of machine learning, aiming to improve model performance by gradually including easy to hard samples in training the model. Self-paced learning (Kumar et al., 2010) measures the difficulty of samples by their training loss, which addressed the issue in previous works that difficulties of samples are generated by prior heuristic rules. Therefore, the model can adjust the curriculum of samples according to its own training status. Following works (Jiang et al., 2015; 2014; Zhou et al., 2020) further proposed many supervised measurement metrics for determining curriculums, for example, the diversity of samples (Jiang et al., 2014) or the consistency of model predictions (Zhou et al., 2020). Meanwhile, many empirical and theoretical studies were proposed to explain why CL could lead to generalization improvement from different perspectives. For example, studies such as MentorNet (Jiang et al., 2018) and Co-teaching (Han et al., 2018) empirically found that utilizing CL strategy can achieve better generalization performance when the given training data are noisy. Gong et al. (2016) provided theoretical explanations on the denoising mechanism that CL learners waste less time with the noisy samples as they are considered harder samples.

Despite great success, most of the existing designed CL strategies are for independent data such as images, and there is little work on generalizing CL strategies to samples with dependencies. Few existing attempts, such as Wang et al. (2021); Chu et al. (2021); Wei et al. (2022); Li et al. (2022), simply treat nodes or graphs as independent samples and then apply CL strategies of independent data, which ignore the fundamental and unique dependency information that carried by the topological structure in data, thus can not well handle the correlation between data samples. Furthermore, these models are mostly based on heuristic-based sample selection strategies (Chu et al., 2021; Wei et al., 2022; Li et al., 2022), which largely limit the generalizability of their proposed methods.

**Graph structure learning.** Another stream of existing studies that are related to our work is *graph structure learning*. Recent studies have shown that GNN models are vulnerable to adversarial attacks on graph structure (Dai et al., 2018; Zügner et al., 2018; Jin et al., 2021; Wu et al., 2019). In order to address this issue, studies in *graph structure learning* usually aim to jointly learn an optimized graph structure and corresponding graph representations. Existing works (Entezari et al., 2020; Chen et al., 2020; Jin et al., 2020; Zheng et al., 2020; Luo et al., 2021) typically consider the hypothesis that the intrinsic graph structure should be sparse or low rank from the original input graph by pruning "irrelevant" edges. Thus, they typically use pre-deterministic methods (Dai et al., 2018; Zügner et al., 2018; Entezari et al., 2020) to preprocess graph structure such as singular value decomposition (SVD), or dynamically remove "redundant" edges according to the downstream task performance on the current sparsified structure (Chen et al., 2020; Jin et al., 2020; Luo et al., 2021). However, modifying the graph topology will inevitably lose potential useful information lying in the removed edges. More importantly, the modified graph structure is usually optimized for maximizing the performance on the training set, which can easily lead to overfitting issues.

## 3 PRELIMINARIES

**Graph Neural Networks** Graph neural networks (GNNs) are a class of methods that have shown promising progress in representing structured data in which data samples are correlated with each other. Typically, the data samples are treated as nodes while their dependencies are treated as edges in the constructed graph. Formally, we denote a graph as $G = (\mathcal{V}, \mathcal{E})$, where $\mathcal{V} = \{v_1, v_2, \ldots, v_N\}$ is a set of nodes that $N = |\mathcal{V}|$ denotes the number of nodes in the graph and $\mathcal{E} \subseteq \mathcal{V} \times \mathcal{V}$ is the set of edges. We also let $\mathbf{X} \in \mathbb{R}^{N \times b}$ denote the node attribute matrix and $\mathbf{A} \in \mathbb{R}^{N \times N}$ represents the adjacency matrix. Specifically, the attribute of node $v_i$ can be expressed as a $b$ dimensional vector $x_i \in \mathbb{R}^b$. $A_{ij} = 1$ denotes there is an edge connecting nodes $v_i$ and $v_j \in \mathcal{V}$, otherwise $A_{ij} = 0$. A GNN model $f$ maps node feature matrix $\mathbf{X}$ associated with the adjacency matrix $\mathbf{A}$ to the model predictions $\hat{\mathbf{y}} = f(\mathbf{X}, \mathbf{A})$, and get the loss $L_{\text{GNN}} = L(\hat{\mathbf{y}}, \mathbf{y})$, where $L$ is the objective function and $\mathbf{y}$ is the ground-truth label of nodes. The loss on one node $v_i$ is denoted as $l_i = L(\hat{y}_i, y_i)$.

**Curriculum Learning** In order to leverage the information carried by the various difficulties of data samples into the training process, Curriculum Learning (CL) (Bengio et al., 2009; Kumar et al., 2010), which is inspired by the cognitive process of human learning principles that learning concepts in a meaningful order (Elman, 1993), is a popular training strategy that can improve the generalization ability of representation learners. Specifically, instead of randomly presenting all training samples to the model as in traditional machine learning algorithms, CL learners start with easy samples and gradually include harder ones during the training process, where the difficulty of samples can be measured by a predetermined policy or a supervised computable metric (e.g. training loss).

## 4 METHODOLOGY

As previous CL strategies have shown that an easy-to-hard learning sequence of independent data samples can improve the generalization ability and robustness of the representation learner, the goal of this paper is to develop an effective CL strategy for handling data with dependencies, which is extremely difficult due to several unique challenges: (1) Difficulty in designing a feasible principle to select edges by properly quantifying their difficulties. (2) Difficulty in designing an appropriate pace of curriculum to gradually involve more edges in training based on model status. (3) Difficulty in ensuring convergence and a numerical steady process for optimizing the model.

In order to address the above challenges, we propose a novel CL method named **R**elational **C**urriculum **L**earning (**RCL**). The sequences, which gradually include edges from easy to hard, are called *curriculums* and learned in different grown-up stages of training. In order to address the first challenge, we propose a joint learning module *Incremental Edge Selection (IES)*, which as shown in Figure 1(a), to select edges that are mostly expected by the current model in a self-supervised manner. Specifically, we jointly learn the node-level prediction task and quantify the difficulty of

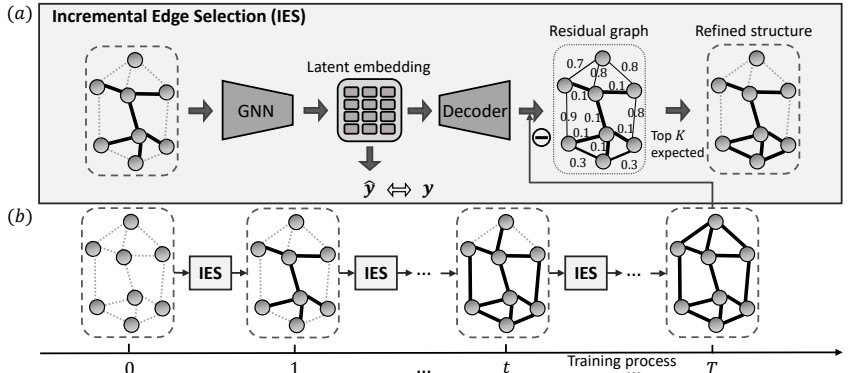

Figure 1: The overall framework of RCL. (a) The *Incremental Edge Selection* module first extracts the latent node embedding by GNN model given the current training structure, then jointly learn the node prediction label **y** and reconstruct the input structure. A small residual error on an edge indicates the corresponding dependency is well expected and thus can be added to the refined structure for the next iteration. (b) The iterative learning process of RCL. Model starts with an empty structure and gradually includes more edges until the training structure converges to the input structure.

edges by their residual error in reconstructing the input structure, then $K$ most expected edges are selected for next iteration, where the details are elaborated in section 4.1. To address the second challenge, which is to design an appropriate pace for gradually including more edges, we present a self-paced learning framework in section 4.2, as shown in Figure 1(b), to automatically increase the number of selected edges $K$ in training given its own training status. Finally, to ensure convergence of optimization, we propose a proximal optimization algorithm with theoretical convergence guarantee in section 4.2 algorithm 1 and we further introduce an edge reweighting scheme to address the numerical issue of discretely incrementing the training structure between iterations in section 4.3.

## 4.1 INCREMENTAL EDGE SELECTION BY QUANTIFYING TOPOLOGY COMPLEXITIES

As mentioned above, we need a novel way to select edges by first quantifying their difficulty levels. However, as existing works on independent data typically use supervised training loss of samples to quantify their difficulty level, there exists no supervised metrics on edges. In order to address this issue, we propose a method to quantify the difficulty of edges by measuring how well the edges are expected from the learned embeddings of their connected nodes. Specifically, the node embedding of the GNN model, which is learned by transforming and aggregating nodes and their neighbors' features through a message-passing fashion, can be considered a high-level semantic representation of data samples and their dependencies. To measure how well the dependencies between data samples are expected, we restore the node embeddings to the original graph structure, which is called the reconstruction of the original graph structure. The residual graph, which is defined as the degree of mismatch between the input adjacency matrix $\mathbf{A}$ and the reconstructed adjacency matrix $\tilde{\mathbf{A}}^{(t)}$, can be considered a strong indicator for describing how well the edges are expected by the current model. Specifically, smaller residual errors indicate a higher probability of being a well-expected edge. In particular, as shown in figure 1(a) that at iteration $t$, we learn the node-level prediction $\hat{\mathbf{y}}^{(t)}$ and extract the latent node embedding $\mathbf{Z}^{(t)}$ from the current GNN model $f^{(t)}$. Then the latent embedding is restored to the original graph structure to reconstruct the input adjacency matrix. The residual graph $\mathbf{R}$ is defined as the degree of mismatch between the input adjacency matrix $\mathbf{A}$ and the reconstructed adjacency matrix $\tilde{\mathbf{A}}^{(t)}$.

With the developed self-supervised method to measure how well edges are expected, here we formulate the key learning paradigm of selecting the top $K$ well-expected edges from the input structure. Specifically, to obtain the training adjacency matrix $\mathbf{A}^{(t)}$ that will be fed into the GNN model $f^{(t)}$, we introduce a learnable binary mask matrix $\mathbf{S}$ with each element $\mathbf{S}_{ij} \in \{0, 1\}$. Thus, the training adjacency matrix at iteration $t$ can be represented as $\mathbf{A}^{(t)} = \mathbf{S}^{(t)} \odot \mathbf{A}$. To filter out the edges with $K$ smallest residual error, we penalize the summarized residual errors over the selected edges, which can be represented as $\sum_{i,j} \mathbf{S}_{ij} \mathbf{R}_{ij}$. Therefore, the learning objective can be presented as follows:

$$\min_{\mathbf{w}} L_{\text{GNN}} + \beta \sum_{i,j} \mathbf{S}_{ij} \mathbf{R}_{ij},$$

$$s.t. \|\mathbf{S}\|_1 \geq K,$$

(1)

where the first term $L_{\text{GNN}} = L(f(\mathbf{X}, \mathbf{A}^{(t)}; \mathbf{w}), \mathbf{y})$ is the node-level predictive loss, e.g. cross-entropy loss for node classification, the second term $\sum_{i,j} \mathbf{S}_{ij} \mathbf{R}_{ij}$ aims at penalizing the residual errors on the edges selected by the mask matrix $\mathbf{S}$ that $\beta$ is a hyperparameter to tune the balance between terms. The constraint is to guarantee only the most $K$ well-expected edges are selected.

Mathematically, the value of a residual edge $\tilde{\mathbf{A}}_{ij}^{(t)} \in [0, 1]$ can be computed by a non-parametric kernel function $\kappa(\mathbf{z}_i^{(t)}, \mathbf{z}_j^{(t)})$, e.g. the inner product kernel. Then the residual error $\mathbf{R}_{ij}$ between the input structure and the reconstructed structure can be defined as $\left\| \tilde{\mathbf{A}}_{ij}^{(t)} - \mathbf{A}_{ij} \right\|$, where $\|\cdot\|$ is commonly chosen to be the squared $\ell_2$-norm. A smaller residual error of edge $\mathbf{A}_{ij}$ denotes that this edge is well-expected, which will be added to the training structure $\mathbf{A}^{(t)}$ at an early stage of training.

## 4.2 SELF-PACED LEARNING OF THE INCREMENTAL STRUCTURE

In order to dynamically include more edges into training, an intuitive way is to iteratively increase the value of $K$ in equation 1 to allow more edges to be selected. However, directly solving equation 1 is difficult since $\mathbf{S}$ is a binary matrix where each element $\mathbf{S}_{ij} \in \{0, 1\}$, optimizing $\mathbf{S}$ would require solving a discrete constraint program at each iteration. Besides, it is difficult to determine an appropriate value of $K$ according to the training status of the model. In order to solve this issue, we first relax the problem into continuous optimization so that each $\mathbf{S}_{ij}$ can be allowed to take any value in the interval $[0, 1]$. Then we treat the constraint as a Lagrange multiplier and solve an equivalent problem by substituting the constraint to a regularization term $g(\mathbf{S}; \lambda)$, thus, our overall loss function can be written as:

$$\min_{\mathbf{w}, \mathbf{S}} L_{\text{GNN}} + \beta \sum_{i,j} \mathbf{S}_{ij} \mathbf{R}_{ij} + g(\mathbf{S}; \lambda), \tag{2}$$

where $g(\mathbf{S}; \lambda) = \lambda \|\mathbf{S} - \mathbf{A}\|$ and $\|\cdot\|$ is commonly chosen to be the squared $\ell_2$-norm. Since the training adjacency matrix $\mathbf{A}^{(t)} = \mathbf{S}^{(t)} \odot \mathbf{A}$, as $\lambda \to \infty$, more edges in the input topology are included until the training adjacency matrix $\mathbf{A}^{(t)}$ converges to the input adjacency matrix $\mathbf{A}$. Specifically, the regularization term $g(\mathbf{S}; \lambda)$ controls the learning scheme by the *age parameter* $\lambda$, where $\lambda = \lambda(t)$ grows with the number of iterations. By monotonously increasing the value of $\lambda$, the regularization term $g(\mathbf{S}; \lambda)$ will push the mask matrix gradually approach the input adjacency matrix $\mathbf{A}$, resulting in more edges involved in the training structure.

**Optimization of learning objective.** It is worth noting that optimizing our objective function in equation 2 requires jointly optimizing parameter $\mathbf{w}$ of GNN model $f$ and the mask matrix $\mathbf{S}$. In order to address this problem, we propose a proximal alternating optimization schema to iteratively update $\mathbf{w}$ and $\mathbf{S}$ in sequence. The full algorithm is presented in Algorithm 1. As we can see, our algorithm takes the input of node feature matrix $\mathbf{X}$ and original adjacency matrix $\mathbf{A}$, a stepsize $\mu$ to control the increasing pace of age parameter $\lambda$, and a hyperparameter $\gamma$ to tune the proximal terms. After initializing the parameters $\mathbf{w}$ and $\mathbf{S}$, it alternates between two updating steps until it finally converges: (1) Step 3 first learns the optimal parameter of GNN model $f$ with the current training adjacency matrix; (2) Step 4&5 extracts the latent node embedding by fixing the GNN model parameter and build the reconstructed adjacency matrix by the kernel function; (3) Step 6 learns the optimal mask matrix $\mathbf{S}$ with the reconstructed adjacency matrix and regularization term; (4) Step 7 refines the training adjacency matrix with respect to the updated mask matrix; (5) The age parameter $\lambda$ is increased when the training adjacency matrix $\mathbf{A}^{(t)}$ is still different from the input adjacency matrix $\mathbf{A}$, thus more edges will be included in the next iteration of the training.

**Theorem 1.** *We have the following convergence guarantees for Algorithm 1:*
*[Avoidance of Saddle Points] If the second derivatives of $L(f(\mathbf{X}, \mathbf{A}^{(t)}; \mathbf{w}), \mathbf{y})$ and $g(\mathbf{S}; \lambda)$ are continuous, then for sufficiently large $\gamma$, any bounded sequence $(\mathbf{w}^{(t)}, \mathbf{S}^{(t)})$ generated by Algorithm1 with random initializations will not converge to a strict saddle point of $F$ almost surely.*
*[Second Order Convergence] If the second derivatives of $L(f(\mathbf{X}, \mathbf{A}^{(t)}; \mathbf{w}), \mathbf{y})$ and $g(\mathbf{S}; \lambda)$ are continuous, and $L(f(\mathbf{X}, \mathbf{A}^{(t)}; \mathbf{w}), \mathbf{y})$ and $g(\mathbf{S}; \lambda)$ satisfy the Kurdyka-Łojasiewicz (KL) property (Wang et al., 2022), then for sufficiently large $\gamma$, any bounded sequence $(\mathbf{w}^{(t)}, \mathbf{S}^{(t)})$ generated by Algorithm 1 with random initialization will almost surely converge to a second-order stationary point of $F$.*

We prove this theorem by Theorem 10 and Corollary 3 from Li et al. (2019) and the complete mathematical proof can be found in Appendix B.

---

**Algorithm 1** Proximal Alternating Minimization Algorithm for Equation 2

---

    **Input** Node features $\mathbf{X}$, adjacency matrix $\mathbf{A}$, a stepsize $\mu$ and hyperparameter $\gamma$
    **Output** The model parameter $\mathbf{w}$ of GNN model $f$.
1: Initialize $\mathbf{w}^{(0)}, \mathbf{S}^{(0)}, \lambda$
2: **while** Not Converged **do**
3:     $\mathbf{w}^{(t)} = \arg\min_{\mathbf{w}} L(f(\mathbf{X}, \mathbf{A}^{(t-1)}; \mathbf{w}), \mathbf{y}) + \beta \sum_{i,j} \mathbf{S}_{ij} \left\| \tilde{\mathbf{A}}_{ij}^{(t-1)} - \mathbf{A}_{ij} \right\| + \frac{\gamma}{2} \left\| \mathbf{w} - \mathbf{w}^{(t-1)} \right\|$
4:     Given $\mathbf{w}^{(t)}$, extract latent nodes embedding $\mathbf{Z}^{(t)}$ from GNN model $f$
5:     Calculate reconstructed structure $\tilde{\mathbf{A}}_{ij}^{(t)} = \kappa(\mathbf{z}_i^{(t)}, \mathbf{z}_j^{(t)})$ for all pairs of $i, j$
6:     $\mathbf{S}^{(t)} = \arg\min_{\mathbf{S}} \beta \sum_{i,j} \mathbf{S}_{ij} \left\| \mathbf{A}_{ij} - \tilde{\mathbf{A}}_{ij}^{(t)} \right\| + g(\mathbf{S}; \lambda) + \frac{\gamma}{2} \left\| \mathbf{S} - \mathbf{S}^{(t-1)} \right\|$
7:     Compute $\mathbf{A}^{(t)} = \mathbf{S}^{(t)} \odot \mathbf{A}$
8:     **if** $\mathbf{A}^{(t)} \neq \mathbf{A}$ **then** increase $\lambda$ by stepsize $\mu$
9: **end while**
10: **return** $\mathbf{w}$

---

### 4.3 SMOOTH TRAINING STRUCTURE BY EDGE REWEIGHTING

Note that in the algorithm 1, the optimization process requires iteratively updating the parameters $\mathbf{w}$ of the GNN model $f$ with respect to the training adjacency matrix $\mathbf{A}^{(t)}$, where the training adjacency matrix $\mathbf{A}^{(t)}$ varies discretely due to the inherent discrete nature of graph topology. However, since state-of-the-art GNN models mostly work in a message-passing fashion, which computes node representations along edges by recursively aggregating and transforming information from neighboring nodes, discretely modifying the number of edges will result in a great drift of the optimal model parameters between iterations. Therefore, the numerical issue caused by discrete changes in the training structure can increase the difficulty of finding optimal parameters and sometimes even hurt the generalization ability of the model. Besides the numerical problem caused by discretely increasing the number of edges, another problem raised by the CL strategy in Section 4.1 is the trustworthiness of the estimated edge difficulty. Recall that the estimated difficulty is inferred from the residual error on the edges, where the reconstructed structure is built based on the learned latent node embedding. Although the residual error can reflect how well edges are expected in the ideal case, the quality of the learned latent node embeddings may affect the validity of this metric and also compromise the quality of the designed curriculum by CL strategy.

To address both issues, we adopt an edge reweighting scheme to (1) smooth the transition of the training structure between iterations, and (2) reduce the influence of edges that connect nodes with relatively low confidence latent embeddings. Formally, we use a smoothed version of structure $\bar{\mathbf{A}}^{(t)}$ to substitute $\mathbf{A}^{(t)}$ for training the GNN model $f$ in step 5 of algorithm 1, where the mapping from $\mathbf{A}^{(t)}$ to $\bar{\mathbf{A}}^{(t)}$ can be represented as:

$$\bar{\mathbf{A}}_{ij}^{(t)} = \beta_{ij}^{(t)} \mathbf{A}_{ij}^{(t)}, \tag{3}$$

where $\beta_{ij}^{(t)}$ is the weight imposed to edge $e_{ij}$ at iteration $t$, and is calculated by considering the counted occurrences of edge $e_{ij}$ until the iteration $t$ and the confidence of the latent embedding for the connected pair of nodes $v_i$ and $v_j$:

$$\beta_{ij}^{(t)} = \rho(v_i)\rho(v_j)\psi(e_{ij}), \tag{4}$$

where $\psi$ is a function that reflects the number of edge occurrences and $\rho$ is a function to reflect the degree of confidence for the latent node embedding. The details of these two functions are as follow.
**Smooth the transition of the training structure between iterations.** In order to obtain a smooth transition of the training structure between iterations, we take the learned curriculum on selected edges by the CL strategy into consideration. Formally, we model $\psi$ by a smooth function of the edge selected times compared to the model iteration times before the current iteration:

$$\psi(e_{ij}) = t(e_{ij})/t, \tag{5}$$

where $t$ is the number of current iterations and $t(e_{ij})$ represents the counting number of selecting edge $e_{ij}$. Therefore, we transform the original discretely changing training structure into a smoothly changing one by taking the historical edge selection curriculum into consideration.
**Reduce the influence of nodes with low confidence latent embeddings.** As introduced in our algorithm 1 line 6, the estimated structure $\tilde{A}$ is inferred from the latent embedding $\mathbf{Z}$ that is extracted from the trained GNN model $f$. Such estimated latent embedding may possibly shift from the true underlying embedding, which results in the inaccurately reconstructed structure around the node.

In order to alleviate this issue, similar to previous CL strategies on inferring the complexity of data samples by their supervised training loss, we model the function $\rho$ by the training loss on nodes, which indicates the confidence of their latent embedding. Specifically, a larger training loss indicates a low confident latent node embedding. Mathematically, the weights $\rho(v_i)$ on node $v_i$ can be represented as a distribution of their training loss:

$$\rho(v_i) \sim e^{-l_i} \tag{6}$$

where $l_i$ is the training loss on node $v_i$. Therefore, a node with a larger training loss will result in a smaller value of $\rho(v_i)$, which reduces the weight of the edges connected to that node.

## 5 EXPERIMENTS

In this section, the experimental settings are introduced first in Section 5.1, then the performance of the proposed method on both synthetic and real-world datasets are presented in Section 5.2. We further present the robustness test on our CL method against topological structure adversarial attack in Section 5.3. Intuitive visualizations of the edge selection curriculum are shown in Section 5.4. In addition, we verify the effectiveness of all components of the method through ablation studies and measure the parameter sensitivity of our proposed method in Appendix A.2 due to the space limit.

### 5.1 EXPERIMENTAL SETTINGS

**Synthetic datasets**  To evaluate the effectiveness of our proposed method on datasets with ground-truth difficulty labels on structure, we follow previous studies (Karimi et al., 2018; Abu-El-Haija et al., 2019) to generate a set of synthetic datasets, where the formation probability of an edge is designed to reflect its likelihood to positively contribute to the node classification job, which indicates its difficulty level. Specifically, the nodes in each generated graph are divided into 10 equally sized node classes $1, 2, \ldots, 10$, and the node features are sampled from overlapping multi-Gaussian distributions. Each generated graph is associated with a *homophily coefficient (homo)* which indicates the probability of a node forming an edge to another node with the same label. For the rest edges that are formed between nodes with different labels, the probability of forming an edge is inversely proportional to the distances between their labels. Therefore, the probability of forming an edge in the synthetic graph can reflect the likelihood that this edge can contribute to the prediction. Specifically, edges with a higher formation probability, e.g. connecting nodes with close labels, meaning that there is a higher chance that this connection will positively contribute to the prediction. We vary the value of *homo* from $0.1, 0.2, \ldots, 0.9$ to generate nine graphs in total. More details and a visualization illustrating the synthetic dataset can be found in Appendix A.1.

**Real-world datasets.**  To further evaluate the performance of our proposed method in real-world scenarios, nine benchmark real-world attributed network datasets, including four citation network datasets Cora, Citeseer, Pubmed (Yang et al., 2016) and ogbn-arxiv Hu et al. (2020), two coauthor network datasets CS and Physics (Shchur et al., 2018), two Amazon co-purchase network datasets Photo and Computers (Shchur et al., 2018), and one protein interation network Hu et al. (2020). We follow the data splits from Chen et al. (2018) on citation networks and use a 5-fold cross-validation

| Homo ratio | 0.1 | 0.2 | 0.3 | 0.4 | 0.5 | 0.6 | 0.7 | 0.8 | 0.9 |
|---|---|---|---|---|---|---|---|---|---|
| GCN | 50.84±1.03 | 56.50±0.50 | 65.17±0.48 | 77.94±0.54 | 87.15±0.44 | 93.27±0.24 | 97.48±0.25 | 99.10±0.17 | 99.93±0.03 |
| GNNSVD | 54.96±0.76 | 58.45±0.56 | 63.06±0.63 | 70.23±0.61 | 80.51±0.41 | 85.02±0.46 | 90.31±0.27 | 94.23±0.22 | 96.74±0.23 |
| ProGNN | 47.87±0.87 | 54.59±0.55 | 65.39±0.44 | 76.96±0.49 | 87.76±0.51 | 93.16±0.34 | 97.60±0.31 | 99.04±0.19 | **99.94±0.03** |
| NeuralSparse | 51.42±1.35 | 57.99±0.69 | 65.10±0.43 | 75.37±0.34 | 87.40±0.29 | 93.54±0.28 | 97.16±0.15 | 99.01±0.22 | 99.83±0.07 |
| PTDNet | 48.21±1.98 | 55.52±2.82 | 65.82±0.94 | 79.37±0.45 | 89.17±0.39 | 94.19±0.18 | 98.61±0.12 | 99.51±0.09 | 99.81±0.05 |
| CLNodes | 50.37±0.73 | 56.64±0.56 | 65.04±0.66 | 77.52±0.48 | 86.85±0.44 | 93.10±0.47 | 97.34±0.25 | 99.02±0.18 | 99.88±0.04 |
| RCL | **57.57±0.43** | **62.06±0.28** | **73.98±0.55** | **84.54±0.75** | **92.69±0.09** | **97.42±0.17** | **99.62±0.05** | **99.89±0.02** | 99.93±0.06 |
| GIN | 48.33±1.89 | 53.62±1.39 | 64.08±0.99 | 77.55±1.10 | 85.31±0.75 | 90.57±0.36 | 97.82±0.18 | 99.59±0.11 | 99.91±0.02 |
| GNNSVD | 43.21±1.60 | 45.68±1.66 | 54.90±1.16 | 68.29±0.79 | 79.76±0.52 | 85.63±0.44 | 93.65±0.39 | 97.22±0.17 | 98.94±0.17 |
| ProGNN | 45.76±1.40 | 52.96±1.01 | 64.12±1.07 | 76.95±0.47 | 85.13±0.71 | 89.96±0.55 | 96.54±0.48 | 99.51±0.12 | 99.78±0.05 |
| NeuralSparse | 50.23±2.05 | 54.12±1.52 | 62.81±0.75 | 76.98±1.17 | 85.14±0.94 | 92.57±0.44 | 98.02±0.20 | 99.61±0.12 | 99.91±0.05 |
| PTDNet | 53.23±2.76 | 56.12±2.03 | 65.81±1.38 | 77.81±1.02 | 86.14±0.65 | 93.21±0.74 | 97.08±0.41 | 99.51±0.18 | 99.91±0.03 |
| CLNodes | 45.36±1.42 | 51.10±1.15 | 62.53±0.88 | 75.83±1.07 | 87.76±0.90 | 94.25±0.44 | **98.30±0.26** | **99.60±0.09** | **99.92±0.03** |
| RCL | **57.63±0.66** | **62.08±1.17** | **71.02±0.61** | **80.61±0.69** | **88.62±0.43** | **94.88±0.36** | 98.19±0.19 | 99.32±0.08 | 99.89±0.04 |
| GraphSage | 62.57±0.55 | 67.33±0.64 | 71.06±0.74 | 80.88±0.54 | 85.88±0.51 | 91.42±0.37 | 95.26±0.33 | 97.78±0.16 | 99.52±0.13 |
| GNNSVD | 64.42±0.80 | 65.71±0.39 | 67.12±0.58 | 68.47±0.50 | 77.70±0.65 | 82.86±0.50 | 87.81±0.71 | 91.61±0.55 | 95.01±0.50 |
| ProGNN | 58.57±2.09 | 66.75±0.91 | 72.14±0.64 | 81.27±0.44 | 86.89±0.47 | 92.10±0.39 | 95.21±0.30 | 97.51±0.23 | 99.50±0.11 |
| NeuralSparse | 61.70±0.77 | 66.65±0.66 | 70.60±0.79 | 79.65±0.45 | 84.19±0.91 | 91.31±0.54 | 94.86±0.53 | 97.16±0.23 | 99.55±0.19 |
| PTDNet | 65.72±1.08 | 69.25±0.92 | 72.60±0.77 | 79.65±0.45 | 86.54±0.56 | 91.79±0.53 | 96.10±0.58 | 97.98±0.13 | **99.78±0.08** |
| CLNodes | **69.41±0.66** | 70.83±0.58 | 75.51±0.36 | 82.65±0.43 | 87.08±0.56 | 91.58±0.41 | 95.91±0.38 | 98.33±0.26 | 99.57±0.14 |
| RCL | 68.03±0.37 | **71.39±0.51** | **76.99±0.99** | **83.76±0.55** | **88.24±0.30** | **93.34±0.56** | **97.66±0.52** | **98.86±0.28** | 99.64±0.08 |

Table 1: Node classification accuracy on synthetic datasets (%). The best-performing method on each backbone GNN model is highlighted in bold, while the second-best method is underlined. In situations where RCL's performance is not strictly the best among all methods, we can see that almost all methods can achieve a near-perfect performance and RCL is still close to the best methods.

| | Cora | Citeseer | Pubmed | CS | Physics | Photo | Computers | ogbn-arxiv | ogbn-proteins |
|---|---|---|---|---|---|---|---|---|---|
| # nodes | 2,708 | 3,327 | 19,717 | 18,333 | 34,493 | 7,650 | 13,752 | 169,343 | 132,534 |
| # edges | 10,556 | 9,104 | 88,648 | 163,788 | 495,924 | 238,162 | 491,722 | 1,166,243 | 39,561,252 |
| # features | 1,433 | 3,703 | 500 | 6,805 | 8,415 | 745 | 767 | 100 | 8 |
| GCN | 85.74±0.42 | 78.93±0.32 | 87.91±0.09 | 93.03±0.32 | 96.55±0.15 | 93.25±0.70 | 88.09±0.40 | 71.74±0.29 | 72.51±0.35 |
| GNNSVD | 83.24±1.03 | 74.80±0.87 | 88.81±0.38 | 93.79±0.11 | 96.11±0.13 | 89.63±0.73 | 86.49±0.77 | 67.44±0.51 | 66.92±0.64 |
| ProGNN | 85.66±0.61 | 74.78±0.55 | 87.22±0.33 | 94.04±0.19 | 96.75±0.26 | 92.07±0.67 | 88.72±0.59 | - | - |
| NeuralSparse | 85.95±0.98 | 76.24±0.48 | 86.83±0.40 | 92.31±0.47 | 95.56±0.30 | 90.57±0.90 | 88.62±0.83 | - | - |
| PTDNet | 83.84±0.95 | 77.54±0.42 | 87.89±0.08 | 93.60±0.43 | 96.56±0.09 | 88.92±0.87 | 87.52±0.70 | - | - |
| CLNode | 85.67±0.33 | 78.99±0.57 | 89.50±0.28 | 93.83±0.24 | 95.76±0.16 | 93.39±0.83 | 89.28±0.38 | 70.95±0.18 | 71.40±0.32 |
| RCL | **87.15±0.44** | **79.79±0.55** | **89.79±0.12** | **94.66±0.32** | **97.02±0.23** | **94.41±0.76** | **90.23±0.23** | **74.08±0.33** | **75.19±0.26** |
| GIN | 84.43±0.65 | 74.87±0.20 | 85.72±0.40 | 91.48±0.36 | 95.62±0.30 | 93.02±0.91 | 86.94±1.58 | 69.26±0.34 | 74.51±0.32 |
| GNNSVD | 82.23±0.65 | 72.11±0.70 | 88.31±0.15 | 91.40±0.87 | 95.30±0.29 | 89.49±1.11 | 82.66±2.26 | 67.79±0.41 | 70.65±0.53 |
| ProGNN | 85.02±0.41 | 78.12±0.93 | 87.82±0.51 | - | - | 92.23±0.67 | 83.54±1.48 | - | - |
| NeuralSparse | 84.92±0.58 | 75.44±0.87 | 86.11±0.49 | 89.66±0.82 | 95.05±0.57 | 93.28±0.83 | 87.22±0.54 | - | - |
| PTDNet | 83.02±1.01 | 75.00±0.74 | 88.04±0.29 | 91.01±0.21 | 95.57±0.40 | 90.70±0.76 | 87.08±0.65 | - | - |
| CLNode | 83.52±0.77 | 75.82±0.58 | 86.92±0.61 | 91.71±0.41 | 95.75±0.46 | 92.78±0.90 | 85.93±1.53 | 70.58±0.17 | 73.97±0.31 |
| RCL | **86.64±0.39** | 77.60±0.18 | **89.17±0.29** | **93.92±0.27** | **96.75±0.17** | **93.88±0.51** | **89.76±0.19** | **72.55±0.15** | **78.76±0.22** |
| GraphSage | 86.22±0.27 | 77.27±0.23 | 88.50±0.16 | 94.22±0.18 | 96.26±0.34 | 93.82±0.51 | 88.62±0.21 | 71.49±0.27 | 77.68±0.20 |
| GNNSVD | 83.11±0.82 | 73.19±0.49 | 88.42±0.38 | 93.86±0.36 | 95.96±0.12 | 89.31±0.53 | 81.46±1.15 | 69.82±0.34 | 71.82±0.39 |
| ProGNN | 86.23±0.42 | 74.45±0.83 | 88.52±0.45 | - | - | 90.89±0.69 | 89.34±0.54 | - | - |
| NeuralSparse | 84.60±0.52 | 76.32±0.55 | 89.02±0.39 | 93.89±0.58 | 96.67±0.20 | 90.78±1.06 | 84.89±1.47 | - | - |
| PTDNet | 86.03±0.60 | 76.07±0.58 | 86.78±0.45 | 93.78±0.43 | 95.32±0.31 | 92.96±0.87 | 84.89±1.47 | - | - |
| CLNode | 86.60±0.64 | 77.23±0.54 | 88.76±0.57 | 94.13±0.34 | 96.87±0.45 | 93.90±0.42 | 89.57±0.62 | 71.54±0.20 | 78.40±0.41 |
| RCL | **86.90±0.39** | **78.95±0.18** | **90.14±0.43** | **95.05±0.23** | **96.88±0.19** | **95.06±0.52** | **90.47±0.38** | **73.13±0.14** | **79.89±0.35** |

Table 2: Node classification results on real-world datasets (%). The best-performing method on each backbone is highlighted in bold and second-best is underlined. (-) denotes an out-of-memory issue.

setting on coauthor and Amazon co-purchase networks. All data are from Pytorch-geometric library (Fey & Lenssen, 2019) and basic statistics are reported in Table 2.

**Comparison methods.** We incorporate three commonly used GNN models, including GCN (Kipf & Welling, 2017), GraphSAGE (Hamilton et al., 2017), and GIN (Xu et al., 2018), as the baseline model and also the backbone model for RCL. In addition to evaluating our proposed method against the baseline GNNs, we further leverage two categories of state-of-the-art methods in the experiments: (1) We incorporate four graph structure learning methods GNNSVD (Entezari et al., 2020), ProGNN (Jin et al., 2020), NeuralSparse (Zheng et al., 2020), and PTDNet (Luo et al., 2021); (2) We further compare with a curriculum learning method CLNode (Wei et al., 2022) which gradually select nodes in the order of the difficulties defined by a heuristic-based strategy. More details about the comparison methods and model implementation details can be found in Appendix A.1.

## 5.2 Effectiveness analysis on synthetic and real-world datasets.

Table 1 presents the node classification results of the synthetic datasets. We report the average accuracy and standard deviation for each model against the *homo* of generated graphs. From the table, we observe that our proposed method RCL consistently achieves the best or most competitive performance to all the comparison methods over three backbone GNN architectures. Specifically, RCL outperforms the second best method on average by 4.17%, 2.60%, and 1.06% on GCN, GIN, and GraphSage backbones, respectively. More importantly, the proposed RCL method performs significantly better than the second best model when the *homo* of generated graphs is low ($\leq 0.5$), on average by 6.55% on GCN, 4.17% on GIN, and 2.93% on GraphSage backbones. These demonstrate that our proposed RCL method significantly improves the model's capability of learning an effective representation to downstream tasks especially when the complexities vary largely in the data.

We report the experimental results of the real-world datasets in Table 2. The results demonstrate the strength of our proposed method by consistently achieving the best results in all 9 datasets by GCN backbone architecture, all 9 datasets by GraphSage backbone architecture, and 6 out of 9 datasets by GIN backbone architecture. Specifically, our proposed method improved the performance of baseline models on average by 1.86%, 2.83%, and 1.62% over GCN, GIN, and GraphSage, and outperformed the second best models model on average by 1.37%, 2.49%, and 1.22% over the three backbone models, respectively. The results demonstrate that the proposed RCL method consistently improves the performance of GNN models in real-world scenarios.

Our experimental results are statically sound. In 43 out of 48 tasks our method outperforms the second-best performing model with strong statistical significance. Specifically, we have in 30 out of 43 cases with a significance $p < 0.001$, in 8 out of 43 cases with a significance $p < 0.01$, and in 5 out of 43 cases with a significance $p < 0.05$. Such statistical significance results can demonstrate that our proposed method can consistently perform better than the baseline models in all datasets.

## 5.3 Robustness analysis against adversarial topological structure attck

To further examine the robustness of the RCL method on extracting powerful representation from complex data samples, we follow previous works (Jin et al., 2020; Luo et al., 2021) to randomly inject fake edges into real-world graphs. This adversarial attack can be viewed as adding random

noise to the topological structure of graphs. Specifically, we randomly connect $M$ pairs of previously unlinked nodes in the real-world datasets, where the value of $M$ varies from 10% to 100% of the original edges. We then train RCL and all the comparison methods on the attacked graph and evaluate the node classification performance. The results are shown in Figure 2, we can observe that RCL shows strong robustness to adversarial structural attacks by consistently outperforming all compared methods on all datasets. Especially, when the proportion of added noisy edges is large ($> 50\%$), the improvement becomes more significant. For instance, under the extremely noisy ratio at 100%, RCL outperforms the second best model by 4.43% and 2.83% on Cora dataset, and by 6.13%, 3.47% on Citeseer dataset, with GCN and GIN backbone models, respectively.

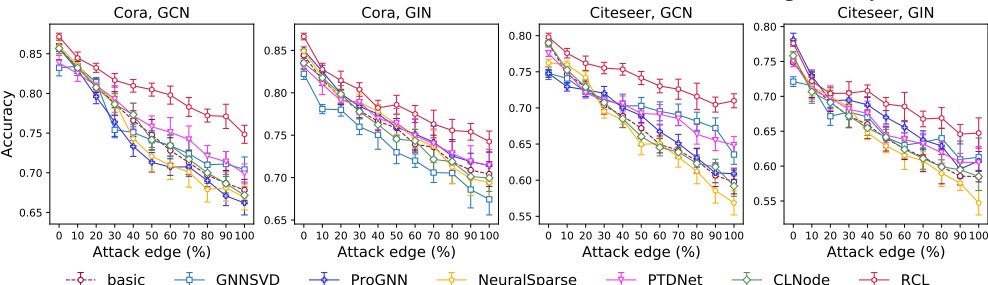

Figure 2: Node classification accuracy (%) on Cora and Citeseer under random structure attack. The attack edge ratio is computed versus the original number of edges, where 100% means that the number of inserted edges is equal to the number of original edges.

## 5.4 VISUALIZATION OF LEARNED EDGE SELECTION CURRICULUM

Besides the effectiveness of the RCL method on downstream classification results, it is also interesting to verify whether the learned edge selection curriculum satisfies the rule from easy to hard. Since real-world datasets do not have ground-truth labels of difficulty on edges, we conduct experiments on synthetic datasets, where the difficulty of each edge can be indicated by its formation probability. Specifically, we classify edges into three categories according to their difficulty: easy, medium, and hard. Here, we define all homogenous edges that connect nodes with the same class as easy, edges connecting nodes with adjacent classes as medium, and the remaining edges connecting nodes with far away classes as hard. We report the proportion of edges selected for each category during training in Figure 3. We can observe that RCL can effectively select most of the easy edges at the beginning of training, then more easy edges and mostly medium edges are gradually included during training, and most hard edges are left unselected until the end stage of training. Such model behavior is consistent with the idea of designing a curriculum for edge selection, which verifies that our proposed method can effectively design curriculums to select edges according to their difficulty from easy to hard, regardless of the backbone model choice.

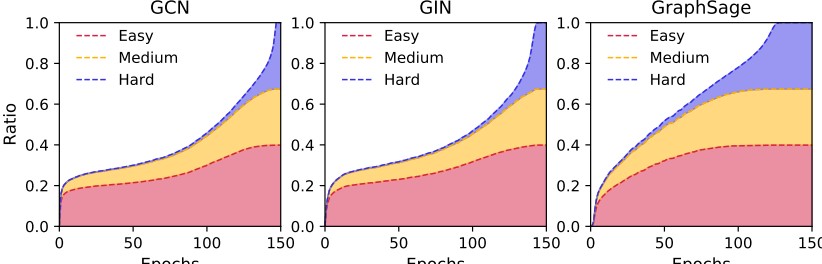

Figure 3: Visualization of edge selection on the synthetic dataset.

## 6 CONCLUSION

This paper focuses on developing a novel CL strategy to improve the generalization ability of GNN models on data samples with dependencies. The proposed method **R**elational **C**urriculum **L**earning (**RCL**) effectively addresses the unique challenges in designing CL strategy for handling dependencies. First, a self-supervised learning module is developed to select appropriate edges that are expected by the model. Then an optimization model is presented to iteratively increment the training structure according to the model training status and a theoretical guarantee of the convergence on the optimization algorithm is given. Finally, an edge reweighting scheme is proposed to steady the numerical process by smoothing the training structure transition. Extensive experiments on synthetic and real-world datasets demonstrate the strength of RCL in improving the generalization ability.

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

## A  ADDITIONAL EXPERIMENTAL SETTINGS AND RESULTS

### A.1  ADDITIONAL EXPERIMENTAL SETTINGS

**Synthetic datasets**    To evaluate the effectiveness of our proposed method on datasets with ground-truth difficulty labels on structure, we first follow previous studies (Karimi et al., 2018; Abu-El-Haija et al., 2019) to generate a set of synthetic datasets, where the difficulty of edges in generated graphs are indicated by their formation probability. Specifically, as shown in Figure 4, each generated graph is with 5,000 nodes, which are divided into 10 equally sized node classes $1, 2, \ldots, 10$. The node features are sampled from overlapping multi-Gaussian distributions. Each generated graph is associated with a *homophily coefficient (homo)* which indicates the likelihood of a node forming a connection to another node with the same label (same color in Figure 4). For example, a generated graph with *homo* $= 0.5$ will have on average half of the edges formed between nodes with the same label. For the rest edges that are formed between nodes with different labels (different colors in Figure 4), the probability of forming an edge is inversely proportional to the distances between their labels. Mathematically, the probability of forming an edge between node $u$ and node $v$ follows $p_{u \to v} \propto e^{-|c_u - c_v|}$, where the distances between labels $|c_u - c_v|$ means shortest distance of two classes on a circle. Therefore, the probability of forming an edge in the synthetic graph can reflect how well this edge is expected. Specifically, edges with a higher formation probability, e.g. connecting nodes with the same label or close labels, meaning that there is a higher chance that this connection will positively contribute to the prediction (less chance to be a noisy edge). Conversely, edges with a lower formation probability, e.g., connecting nodes with faraway labels, mean that there is a higher chance that this connection will negatively contribute to the prediction (higher chance to be a noisy edge). We vary the value of *homo* from $0.1, 0.2, \ldots, 0.9$ to generate nine graphs in total. Similar to previous works (Karimi et al., 2018; Abu-El-Haija et al., 2019), we randomly partition each synthetic graph into equal-sized train, validation, and test node splits.

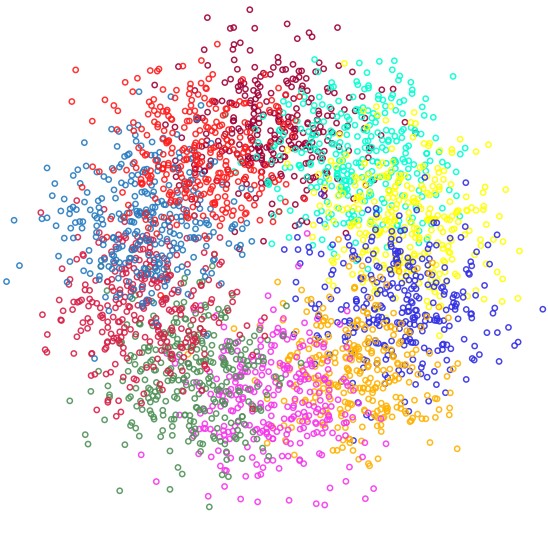

Figure 4: Visualization of synthetic datasets. Each color represents a class of nodes. Node attributes are sampled from overlapping multi-Gaussian distributions, where the attributes of nodes with close labels are likely to have short distances. Homogeneous edges represent edges that connect nodes of the same class (with the same color). The probability of connecting two nodes of different classes decreases with the distance between the center points of their class distribution. Therefore, the formation probability of a node denotes the edge difficulty, since edges between nodes with close classes are more likely to positively contribute to the prediction under the homogeneous assumption.

**Initializing graph structure by a pre-trained model.**    It is worth noting that the model needs an initial training graph structure $\mathbf{A}^{(0)}$ in the initial stage of training. An intuitive way is that we can initialize the model to work in a purely data-driven scenario that starts only with isolated nodes where no edges exist. However, an instructive initial structure can greatly reduce the search cost

and computational burden. Inspired by many previous CL works (Weinshall et al., 2018; Hacohen & Weinshall, 2019; Jiang et al., 2018; Zhou et al., 2020) that incorporate prior knowledge of a pre-trained model into designing curriculum for the current model, we initialize the training structure $\mathbf{A}^{(0)}$ by a pre-trained vanilla GNN model $f^*$. Specifically, we follow the same steps from line 4 to line 7 in the algorithm 1 to obtain the initial training structure $\mathbf{A}^{(0)}$ but the latent node embedding is extracted from the pre-trained model $f^*$.

**Implementation Details**  We use the baseline model (GCN, GIN, GraphSage) as the backbone model for both our RCL method and all comparison methods. For a fair comparison, we require all models follow the same GNN architecture with two convolution layers. For each split, we run each model 10 times to reduce the variance in particular data splits. Test results are according to the best validation results. General training hyperparameters (such as learning rate or the number of training epochs) are equal for all models. For the pre-trained model to initialize the training structure, we utilize the same model as the backbone model utilized by our method. For example, if we use GCN as the backbone model for RCL, the pre-trained model to initialize is also GCN. All experiments are conducted on a 64-bit machine with four NVIDIA Quadro RTX 8000 GPUs. The proposed method is implemented with Pytorch deep learning framework (Paszke et al., 2019).

The following describes the details of our comparison models.

**Graph Neural Networks (GNNs).** We first introduce three baseline GNN models as follows.

**(i) GCN.** Graph Convolutional Networks (GCN) (Kipf & Welling, 2017) is a commonly used GNN, which introduces a first-order approximation architecture of the Chebyshev spectral convolution operator;

**(ii) GIN.** Graph Isomorphism Networks (GIN) (Xu et al., 2018) is a variant of GNN, which has provably powerful discriminating power among the class of 1-order GNNs;

**(iii) GraphSage.** GraphSage Hamilton et al. (2017) is a GNN method that computes the hidden representation of the root node by aggregating the hidden node representations hierarchically from bottom to top.

**Graph structure learning.** We then introduce four state-of-the-art methods for jointly learning the optimal graph structure and downstream tasks.

**(i) GNNSVD.** GNNSVD (Entezari et al., 2020) first apply singular value decomposition (SVD) on the graph adjacency matrix to obtain a low-rank graph structure and apply GNN on the obtained low-rank structure;

**(ii) ProGNN.** ProGNN (Jin et al., 2020) is a method to defend against graph adversarial attacks by obtaining a sparse and low-rank graph structure from the input structure;

**(iii) NeuralSparse.** NeuralSparse (Zheng et al., 2020) is a method to learn robust graph representations by iteratively sampling $k$-neighbor subgraphs for each node and sparsing the graph according to the performance on the node classification;

**(iv) PTDNet.** PTDNet (Luo et al., 2021) learns a sparsified graph by pruning task-irrelevant edges, where sparsity is controlled by regulating the number of edges.

**Curriculum learning on graph data.** We introduce a recent curriculum learning work on node classification as follows.

**(i) CLNode.** CLNode (Wei et al., 2022) regards nodes as data samples and gradually incorporates more nodes into training according to their difficulty. They apply a heuristic-based strategy to measure the difficulty of nodes, where the nodes that connect neighboring nodes with different classes are considered difficult.

**Searching space for hyperparameters.**
Number of epochs trained: $\{150, 500\}$;
Learning rate for model: $\{1e-2, 5e-3, 1e-3\}$;
Number of GNN layers: $\{2\}$;
Dimension of hidden state: $\{64\}$;
Age parameter $\lambda$ : $\{1, 2, 3, 4, 5\}$ (A larger value indicates faster pacing for adding edges, where 1 denotes the training structure will converge to the input structure at the final iteration).

## A.2 ADDITIONAL EXPERIMENTS

|  | Synthetic1 | Synthetic2 | Citeseer | CS | Computers |
|---|---|---|---|---|---|
| Full | **73.98±0.55** | **97.42±0.17** | **79.79±0.55** | **94.66±0.22** | **90.23±0.23** |
| w/o edge appearance | 70.70±0.43 | 95.77±0.16 | 77.77±0.65 | 94.39±0.21 | 89.56±0.30 |
| w/o node confidence | 72.38±0.41 | 96.86±0.17 | 78.72±0.72 | 94.34±0.13 | 90.03±0.62 |
| w/o pre-trained model | 72.56±0.69 | 93.89±0.14 | 78.28±0.77 | 94.50±0.14 | 89.80±0.55 |

Table 3: Ablation study. Here "Full" represents the original method without removing any component. The best-performing method on each dataset is highlighted in bold.

**Ablation study** Here we investigate the impact of the proposed components of RCL. We first consider variants of removing the structural smoothing components mentioned in Section 4.3. Specifically, we consider two variants *w/o edge appearance* and *w/o node confidence*, which remove the smoothing function of the edge appearance ratio and the component to reflect the degree of confidence for the latent node embedding in RCL, respectively. In addition to examining the effectiveness of components in structural smoothing, we further consider a variant *w/o pre-trained model* that avoids using a pre-trained model, which is mentioned in Appendix A.1, to initialize the training structure by a pre-trained model and instead starts with inferred structure from isolated nodes with no connections. We present the results of two synthetic datasets (*homophily coefficient*$= 0.3, 0.6$) and three real-world datasets in Table 3, where we can observe a significant performance drop consistently for all variants. The results validate that all structural smoothing and initialization components can benefit the performance of RCL on the downstream tasks.

**Parameter sensitivity analysis** Recall that RCL learns a curriculum to gradually add edges in a given input graph structure to the training process until all edges are included. An interesting question is how the speed of adding edges will affect the performance of the model. Here we conduct experiments to explore the impact of age parameter $\lambda$ which controls the speed of adding edges to the model performance. Here a larger value of $\lambda$ means that the training structure will converge to the input structure earlier. For example, $\lambda = 1$ means that the training structure will probably not converge to the input structure until the last iteration, and $\lambda = 5$ means that the training structure will converge to the input structure around half of the iterations are complete, and then the model will be trained with the full input structure for the remaining iterations. We present the results on two synthetic datasets (*homophily coefficient*$= 0.3, 0.6$) and two real-world datasets in Figure 5. As can be seen from the figure, the classification results are steady that the average standard deviation is only 0.41%. It is also worth noting that the peak values for all datasets consistently appear around $\lambda = 3$, which indicates that the best performance is when the training structure converges to the full input structure around two-thirds of the iterations are completed.

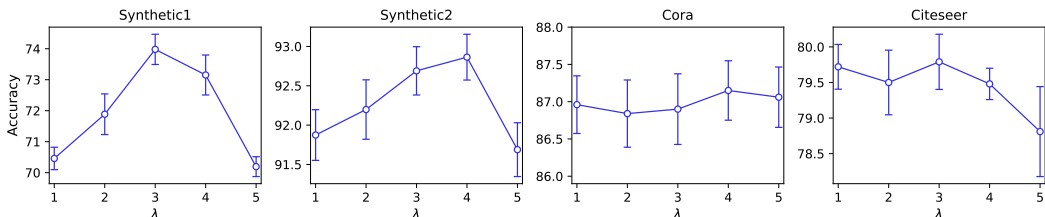

Figure 5: Parameter sensitivity analysis on four datasets. Here a larger value of $\lambda$ means the training structure will converge to the original structure at an earlier training stage.

## B MATHEMATICAL PROOF

**Theorem 1.** We have the following convergence guarantees for Algorithm 1:
**[Avoidance of Saddle Points]** If the second derivatives of $L(f(\mathbf{X}, \mathbf{A}^{(t)}; \mathbf{w}), \mathbf{y})$ and $g(\mathbf{S}; \lambda)$ are continuous, then for sufficiently large $\gamma$, any bounded sequence $(\mathbf{w}^{(t)}, \mathbf{S}^{(t)})$ generated by Algorithm1

with random initializations will not converge to a strict saddle point of $F$ almost surely.

**[Second Order Convergence]** If the second derivatives of $L(f(\mathbf{X}, \mathbf{A}^{(t)}; \mathbf{w}), \mathbf{y})$ and $g(\mathbf{S}; \lambda)$ are continuous, and $L(f(\mathbf{X}, \mathbf{A}^{(t)}; \mathbf{w}), \mathbf{y})$ and $g(\mathbf{S}; \lambda)$ satisfy the Kurdyka-Łojasiewicz (KL) property (Wang et al., 2022), then for sufficiently large $\gamma$, any bounded sequence $(\mathbf{w}^{(t)}, \mathbf{S}^{(t)})$ generated by Algorithm 1 with random initialization will almost surely converges to a second-order stationary point of $F$.

*Proof.* We prove this theorem by Theorem 10 and Corollary 3 from Li et al. (2019).

**[Avoidance of Saddle Points]** Because the sequence $(\mathbf{w}^{(t)}, \mathbf{S}^{(t)})$ is bounded, and the second derivatives of $L$ and $g$ are continuous, then they are bounded. In other words, we have $\max\{\left\|\nabla_{\mathbf{w}}^2 L(f(\mathbf{X}, \mathbf{A}^{(t)}; \mathbf{w}^{(t)}), \mathbf{y})\right\|, \left\|\nabla_{\mathbf{S}}^2 g(S^{(t)}; \lambda)\right\|\} \leq p$, where $p > 0$ is a constant. Similarly, it is easy to check that the second derivative of the term $\sum_{i,j} \mathbf{S}_{ij} \left\|\tilde{\mathbf{A}}_{ij}^{(t)} - \mathbf{A}_{ij}\right\|_2^2$ is bounded, i.e., $\max\{\left\|\nabla_{\mathbf{w}}^2 \sum_{i,j} \mathbf{S}_{ij} \left\|\tilde{\mathbf{A}}_{ij}^{(t)} - \mathbf{A}_{ij}\right\|_2^2\right\|, \left\|\nabla_{\mathbf{S}}^2 \sum_{i,j} \mathbf{S}_{ij} \left\|\tilde{\mathbf{A}}_{ij}^{(t)} - \mathbf{A}_{ij}\right\|_2^2\right\|\} \leq q$, where $q > 0$ is constant and $\tilde{\mathbf{A}}$ is a function of $\mathbf{w}$. Therefore, it means that the objective $F$ is bi-smooth, i.e. $\max\{\left\|\nabla_{\mathbf{w}}^2 F\right\|, \left\|\nabla_{\mathbf{S}}^2 F\right\|\} \leq p + q$. In other words, $F$ satisfies Assumption 4 from Li et al. (2019). Moreover, the second derivative of $F$ is continuous. For any $\gamma > p + q$, any bounded sequence $(\mathbf{w}^{(t)}, \mathbf{S}^{(t)})$ generated by Algorithm 1 will not converge to a strict saddle of $F$ almost surely by Theorem 10 from Li et al. (2019).

**[Second Order Convergence]** From the above proof of avoidance of saddle points, we know that $F$ satisfies Assumption 4 from Li et al. (2019). Moreover, because $L$ and $g$ satisfy the KL property, and the term $\sum_{i,j} \mathbf{S}_{ij} \left\|\tilde{\mathbf{A}}_{ij}^{(t)} - \mathbf{A}_{ij}\right\|_2^2$ satisfies the KL property, we conclude that $F$ satisfy the KL property as well. From the proof above, we also know that the second derivative of $F$ is continuous. Because continuous differentiability implies Lipschitz continuity Wheeden & Zygmund (1977), it infers that the first derivative of $F$ is Lipschitz continuous. As a result, $F$ satisfies Assumption 1 from Li et al. (2019). Because $F$ satisfies Assumptions 1 and 4, then for any $\gamma > p + q$, any bounded sequence $(\mathbf{w}^{(t)}, \mathbf{S}^{(t)})$ generated by Algorithm 1 will almost surely converges to a second-order stationary point of $F$ by Corollary 3 from Li et al. (2019). $\square$

While the convergence of Algorithm 1 entails the second-order optimality conditions of $f$ and $g$, some commonly used $f$ such as the GNN with sigmoid or tanh activations and some commonly used $g$ such as the squared $\ell_2$ norm satisfy the KL property, and Algorithm 1 is guaranteed to avoid a strict saddle point and converges to a second-order stationary point.

