# OpenReview forum: "Relational Curriculum Learning for Graph Neural Networks"
_ICLR.cc/2023/Conference — Submitted to ICLR 2023_

### Official Review · Reviewer_fwhF · 2022-10-25

**Confidence:** 4
**Clarity, Quality, Novelty And Reproducibility:** See the comments above.
**Correctness:** 3
**Technical Novelty And Significance:** 3
**Empirical Novelty And Significance:** 2
**Recommendation:** 6

**Strength And Weaknesses:**

Strengths:
1. The problem of learning graph structures for GNN is important.
2. The proposed method uses the node representations learned by GNNs to measure the likelihood of each edge, which is quite intuitive.
3. Some improvements are observed in the experiment.

Weaknesses:
1. The writing of the paper can be further improved.
2. The improvements over existing methods are not significant.

Detailed comments:

1. The writing of the paper can be further improved.
The abstract of the paper says that a curriculum learning method is proposed, where edges are given to GNNs according to their difficulty from easy to hard, and the difficulty is measured by a self-supervised learning paradigm. However, in the model section, these claims are not well explained. The edge difficulty and self-supervised learning paradigm are not mentioned. Although readers can roughly get the idea that the edge difficulty is measured by the inner product between corresponding node embeddings, I feel like the abstract and model sections are written by different authors, making it not easy to follow.

2. Many ideas of the paper are not new
The overall loss function of the proposed method is given by Eq. (3) of Algorithm 1. There are three parts, where the first part is a supervised loss, the second part is a graph reconstruction loss, and the third one is a regularization term. The first two terms are widely studied in the graph machine learning literature for GNN training and structure learning. Given these existing works, I feel like this paper does not bring new ideas. Although the authors introduce their method from a perspective of curriculum learning and some smoothing strategies are proposed in Sec. 4.3, I think the contribution is not so significant as the underlying ideas are not so different from existing methods.

3. The improvements over existing methods are not very significant.
The results on real-world datasets are presented in Tab. 2, where the improvements over existing methods are not so significant, and the results on different datasets all have very high standard deviation. To better convince readers, I think it is helpful to consider some larger datasets for evaluation, where the standard deviation over different runs can be smaller.

**Summary Of The Paper:**

This paper investigates curriculum learning for graph neural networks (GNNs), aiming at selecting the most important edges in a graph for GNN training. The authors formalize the task as an optimization problem, which has a node-level predictive loss and a structural-penalty loss. A proximal algorithm is proposed for optimization. Experimental results on multiple datasets are encouraging.

**Summary Of The Review:**

See the comments above.

---

> ### Author Response · Authors · 2022-11-11
> **Response to Reviewer fwhF (1/2)**
>
> We thank the reviewer for your detailed and insightful comments.
>
> Review #1.
> The writing of the paper can be further improved. The abstract of the paper says that a curriculum learning method is proposed, where edges are given to GNNs according to their difficulty from easy to hard, and the difficulty is measured by a self-supervised learning paradigm. However, in the model section, these claims are not well explained. The edge difficulty and self-supervised learning paradigm are not mentioned. Although readers can roughly get the idea that the edge difficulty is measured by the inner product between corresponding node embeddings, I feel like the abstract and model sections are written by different authors, making it not easy to follow.
>
> Response #1.
> Thank you for your suggestion and we have revised the model section of our paper for the purpose of clarification. In general, the difficulty level of the edges represents the likelihood that the corresponding connection can positively contribute to the node prediction task, and the self-supervised learning paradigm is introduced to quantify the difficulty levels of edge connections. Previous curriculum learning strategies typically use supervised computable metrics such as the training loss to quantify the difficulty levels of data samples. However, this method can not be easily generalized to quantify the difficulty level of edges, because the training target is on the node level and there exists no supervised computable metrics on edges. To address this challenge, we proposed a self-supervised learning metric to measure the difficulty level by how well the edges are expected by their connecting node embeddings. Specifically, for each edge a kernel function (e.g. inner product kernel) is applied to the learned embeddings of two connecting nodes, where a higher score represents this edge is more expected to be added to the training structure.
>
> Review #2.
> Many ideas of the paper are not new The overall loss function of the proposed method is given by Eq. (3) of Algorithm 1. There are three parts, where the first part is a supervised loss, the second part is a graph reconstruction loss, and the third one is a regularization term. The first two terms are widely studied in the graph machine learning literature for GNN training and structure learning. Given these existing works, I feel like this paper does not bring new ideas. Although the authors introduce their method from a perspective of curriculum learning and some smoothing strategies are proposed in Sec. 4.3, I think the contribution is not so significant as the underlying ideas are not so different from existing methods.
>
> Response #2.
> 1. Our overall loss function of the proposed method is given by Equation 2 in Section 4.2 instead of Equation (3) of Algorithm 1, where Equation (3) of Algorithm 1 is only one step in our alternating optimization algorithm.
>
> 2. One contribution of our work is that we proposed a novel curriculum learning strategy to handle dependent data by gradually incorporating edges into the training of GNN models according to their difficulty level. In order to quantify the difficulty level for edge selection, we introduced measuring the residual error in the reconstructed graph structure. However, our edge selection process is different from a standard graph reconstruction process, where the second term of our overall loss function is different from the commonly used graph reconstruction loss. Instead of evenly penalizing the reconstruction loss on all edges, it aims at penalizing the reconstruction loss over the edges selected by a binary mask matrix $\mathbf{S}$. Therefore, our model can effectively select the edges with relatively small reconstruction errors, which means these edges are more expected to be added to the training structure.
>
> 3. Another novel contribution of our work is to design an appropriate pacing function to gradually involve edges based on the model status. Previous works typically use a fixed pacing function to control the selection process, which may lead to suboptimal results. In order to add appropriate edges that are ready for the current model, we proposed the third term $g(\mathbf{S};\lambda) = \lambda \left\| \mathbf{S} - \mathbf{A} \right\|$ in our overall loss function, which is not a trivial regularization term because it controls the learning scheme by the \textit{age parameter} $\lambda$, where $\lambda=\lambda(t)$ grows with the number of iterations. By monotonously increasing the value of $\lambda$, this regularization term $\lambda \left\| \mathbf{S} - \mathbf{A} \right\|$ will push the mask matrix $\mathbf{S}$ gradually approach the input adjacency matrix $\mathbf{A}$, resulting in more edges involved in the training structure based on the current model status.

---

> > ### Author Response · Authors · 2022-11-11
> > **Response to Reviewer fwhF (2/2)**
> >
> > Review #3.
> > The improvements over existing methods are not very significant. The results on real-world datasets are presented in Tab. 2, where the improvements over existing methods are not so significant, and the results on different datasets all have very high standard deviation. To better convince readers, I think it is helpful to consider some larger datasets for evaluation, where the standard deviation over different runs can be smaller.
> >
> > Response #3.
> > Thank you for your suggestions, we have added additional experiments on two large graph datasets and statistical significance analysis on the experimental results.
> >
> > 1. We use ogbn-proteins and ogbn-arxiv datasets from the Open Graph Benchmark [1] for additional experiments, where ogbn-proteins has 132,534 nodes and 39,561,252 edges, and ogbn-arxiv has 169,343 nodes and 1,166,243 edges. We follow the original train/validation/test split settings and report the accuracy/ROCAUC score on ogbn-arxiv and ogbn-proteins respectively according to the Open Graph Benchmark [1]. All models follow the same architecture with three convolution layers and the hidden dimension is 256. The averaged experimental results over 10 times running are shown below:
> >
> > -------------------------------------------------------------------------------------------------------------------------
> >             | GCN       | GNNSVD+GCN | CLNode + GCN| Ours + GCN
> >
> > ogbn-arxiv     | 71.74$\pm$0.29         | 67.44$\pm$0.51              | 70.95$\pm$0.18                | 74.08$\pm$0.33
> >
> > ogbn-proteins | 72.51$\pm$0.35 | 66.92$\pm$0.64       | 71.40$\pm$0.32         | 75.19$\pm$0.26
> >
> >             | GIN       | GNNSVD+GIN | CLNode + GIN| Ours + GIN
> >
> > ogbn-arxiv    | 69.26$\pm$0.34 | 67.79$\pm$0.41       | 70.58$\pm$0.17         | 72.55$\pm$0.15
> >
> > ogbn-proteins | 74.51$\pm$0.32 | 70.65$\pm$0.53       | 73.97$\pm$0.31         | 78.76$\pm$0.22
> >
> >            | GraphSage  | GNNSVD+GraphSage| CLNode + GraphSage| Ours + GraphSage
> >
> > ogbn-arxiv    | 71.49$\pm$0.27 | 69.82$\pm$0.34       | 71.54$\pm$0.20         | 73.13$\pm$0.14
> >
> > ogbn-proteins | 77.68$\pm$0.20 | 71.82$\pm$0.39       | 78.40$\pm$0.41         | 79.89$\pm$0.35
> >
> > -------------------------------------------------------------------------------------------------------------------------
> >
> > We can observe that our method consistently outperforms all baseline models on two datasets with three commonly used backbone GNN models. Specifically, our proposed method outperforms the second-best-performing model by 2.21\% on average. Besides, in 5 out of 6 cases, we have the statistical significance score $p<0.001$, and we have $p<0.01$ for the left case. It is also worth noting that other comparison methods used in our paper such as ProGNN, NeuralSparse, and PTDNet can not handle these two large datasets and went into an out-of-memory error.
> >
> > 2. Our original experimental results in the paper are statistically sound. We further added the statistical significance analysis to the experimental results in the revised paper. Our proposed method has outperformed all baseline models in 41 out of 48 tasks on synthetic and real-world datasets, where in 37 out of 41 cases our method outperforms the second-best performing model with strong statistical significance. Specifically, we have in 25 out of 37 cases with a significance $p<0.001$, in 7 out of 37 cases with a significance $p<0.01$, and in 5 out of 37 cases with a significance $p<0.05$. Such statistical significance results can demonstrate that our proposed method can consistently perform better than the baseline models in all datasets.
> >
> > [1] Hu, Weihua, et al. "Open graph benchmark: Datasets for machine learning on graphs." Advances in neural information processing systems 33 (2020): 22118-22133.

---

> > > ### Comment · Reviewer_fwhF · 2022-11-24
> > > **Response**
> > >
> > > Thanks the authors for the feedback, including some helpful discussions and more experimental results. The feedback addressed some of my concerns on the model performance, so I will raise my score to a weak accept.

---

### Official Review · Reviewer_f5Ws · 2022-10-25

**Confidence:** 4
**Correctness:** 3
**Technical Novelty And Significance:** 3
**Empirical Novelty And Significance:** 3
**Recommendation:** 6

**Clarity, Quality, Novelty And Reproducibility:**

The paper is a nice addition to the growing literature of GNNs, emphasizing on structure learning. Overall it is a readable paper, although added concepts like curriculum learning, self-paced learning and self-supervised learning make the messages unclear.

**Strength And Weaknesses:**

Strength
======
- The paper studies an interesting problem -- optimizing the parameter and graph structure jointly.
- The proposed solution is sensible and is shown empirically to improve performance and robustness.

Weaknesses
==========
- Posing the forward edge selection as curriculum learning is somewhat confusing. It is better to emphasize on the joint optimization aspects.
- Again, adding terms like self-paced, self-supervised learning are also confusing.
- Although homophily is a well-known properties of real-world graphs, it would be interesting to see if this assumption breaks, e.g., redesigning the kernel for graph reconstruction.

**Summary Of The Paper:**

The paper introduces a method for iterative construction of graph structure in GNN settings. The key is to make use of the homophily -- similar nodes tend to connect. The optimization procedures encourage more edges to be added along the training process. This results in a better optimized GNNs -- boosting up accuracy performance and improved robustness against structural noise.

**Summary Of The Review:**

An useful attention to the literature of graph structure learning in GNNs. I'd like to see when the homophily assumption breaks and how the method will be adjusted.

---

> ### Author Response · Authors · 2022-11-11
> **Response to Reviewer f5Ws**
>
> Thank you very much for your consideration and insightful comments.
>
> Review #1.
> Posing the forward edge selection as curriculum learning is somewhat confusing. It is better to emphasize on the joint optimization aspects.
>
> Response #1.
> In general, the idea of curriculum learning motivated our work to gradually select more edges in GNN training for a better model generalization ability. Therefore, to propose a feasible end-to-end framework, we formalize the problem as a joint optimization learning problem of node-level prediction and edge selection.
>
> Specifically, curriculum learning, as we introduced in Section 1 Introduction, and Section 2 Related Works, is a learning strategy that was widely used to mimic the human learning principles to learn concepts in a meaningful order. Many previous studies have found that such an easy-to-hard learning strategy on data samples can improve the generalization ability of machine learning algorithms. Inspired by these previous works, we generalize the idea of curriculum learning to further handle data dependencies, which can be treated as edges in the graph data. To simultaneously learn node-level objectives and select appropriate edges given the current model state, we formalize the entire learning process as a joint optimization problem that can iteratively update node-level predictions and add edges during training.
>
>
> Review #2. Again, adding terms like self-paced, self-supervised learning are also confusing.
>
> Response #2.
> 1. Self-paced learning is to give the model a certain degree of freedom to adjust the edge selection pace according to its current training status. Specifically, many previous curriculum learning strategies use a pre-defined fixed pacing function of the training iterations (e.g. exponential pacing function). However, such simple strategies do not give enough freedom for the model to select the optimal edges given the current training status and can lead to suboptimal performance. In this paper, we introduced self-paced learning as a dynamic regularization term $g$ in Equation 2 of Section 4.2 to automatically involve more edges in training, where the model retains a certain degree of freedom to adapt the pacing of adding edges according to its current training status.
> 2. An unique challenge to designing an effective curriculum learning strategy on dependent data samples is how to properly quantify the difficulty levels of edges. As we introduced in Paragraph 4 of Section 1, previous curriculum learning studies typically use supervised computable metrics, for example, the training loss, to quantify the difficulty levels of data samples. However, this method can not be easily generalized to quantify the difficulty level of edges, because the training target is on the node level and there exists no supervised metrics on edges. To address this challenge, we proposed a self-supervised learning metric to measure how well the edges are expected by their connecting node embeddings. Specifically, the difficulty level of an edge is quantified by a kernel function of its two connecting nodes' embedding, where a higher score denotes this connection is more expected to be added to the training structure.
>
>
> Review #3.
> Although homophily is a well-known properties of real-world graphs, it would be interesting to see if this assumption breaks, e.g., redesigning the kernel for graph reconstruction.
>
> Response #3. We would like to clarify that our proposed method does not require a homophily assumption, where our method is a general learning framework that any non-parametric kernels can be utilized in determining how well an edge is expected by its connecting nodes embeddings. Our proposed learning framework can support other possible choices of kernels, such as a tanh kernel or polynomial kernel, without any further modifications to the learning framework. The inner-product kernel was used in our experiments due to its simplicity and consistent good experimental performance compared to baseline models on both synthetic datasets and real-world datasets. We sincerely thank the reviewer for pointing out that investigating the design of a kernel that breaks the homophily assumption can be a very interesting research question. However, since the goal of this paper is to design a feasible curriculum learning strategy on gradually involving edges in training GNN models to improve the generalization ability, investigating the specific choice of kernel function goes beyond the scope of this work. But we would consider this as a promising future work.
>
>
> Review #4. An useful attention to the literature of graph structure learning in GNNs. I'd like to see when the homophily assumption breaks and how the method will be adjusted.
>
> Response #4. Thank you for your precious suggestion, as mentioned above, we would definitely consider this as our future work.

---

> > ### Author Response · Authors · 2022-12-06
> > **A warm reminder**
> >
> > Dear Reviewer,
> >
> > Thanks again for your time.
> >
> > As the end date of Discussion Stage 2 is approaching, we hope that all your concerns have been addressed. If any concerns remain, we are happy to further clarify them. Looking forward to your reply.
> >
> > Best regards,
> >
> > Authors

---

### Official Review · Reviewer_yJ3m · 2022-10-26

**Confidence:** 3
**Correctness:** 2
**Technical Novelty And Significance:** 3
**Empirical Novelty And Significance:** 2
**Recommendation:** 5

**Clarity, Quality, Novelty And Reproducibility:**

The clarity is OK.

The quality is questionable since some key points are not clear.

Novelty is good.

Reproducibility is unclear, and can only be checked by running the code.


**Details Of Ethics Concerns:**

No.

**Strength And Weaknesses:**


Strengths:

1. The paper is overall easy to understand. The motivation and the background knowledge of the curriculum learning are clearly introduced.

2. The idea to achieve better learning performance on graphs by investigating the difficulty of the edges is novel.

3. In experiments, both synthetic datasets and real-world datasets are used. The synthetic datasets have controllable properties (i.e. homophily coefficient).


Weakness:

1. The main idea is pretty intuitive. The difficulty level of the edges is only intuitively proposed without concrete examples. Therefore, it is unclear what exactly the difficulty levels mean. Moreover, the noise seems to be one component of the difficulty, is there any other concrete factors determining the difficulty?

2. i.d.d data is not correctly used. What the authors want to express through i.d.d data should be 'independent data', in which the data have no connections. While i.d.d. data refers to different samples drawn independently from identical distributions.

3. In experiments, it is unclear why the difficulty of the edges can correspond to the formation probability.

4. The datasets are rather small.

5. The proposed method is not consistently better than the baselines in all datasets. Besides, is there any experiment that can demonstrate that it is indeed the curriculum learning part that improves the performance, instead of other factors like model complexity?


**Summary Of The Paper:**

This paper targets the problem that the edges in a graph may have different levels of difficulty to learn and proposes a curriculum learning-based model to gradually incorporate edges during learning according to their difficulties. The difficulty level is obtained by self-supervised learning. Experiments are conducted on 9 synthetic datasets (with 9 different homophily coefficients) and 7 real-world datasets.

**Summary Of The Review:**

Overall, this paper has a novel idea, but the proposed model is not totally clear. Besides, misusing the concepts like i.d.d. makes the paper not rigorous enough.

---

> ### Author Response · Authors · 2022-11-11
> **Response to Reviewer yJ3m (1/4)**
>
> Thank you very much for your summarization and insightful comments.
>
> Review #1. The main idea is pretty intuitive. The difficulty level of the edges is only intuitively proposed without concrete examples. Therefore, it is unclear what exactly the difficulty levels mean. Moreover, the noise seems to be one component of the difficulty, is there any other concrete factors determining the difficulty?
>
> Response #1. The difficulty level of the edges can be understood as the likelihood that the corresponding data connecting mechanism can positively contribute to the downstream prediction task. Therefore, the underlying reason that determines the difficulty level can vary due to many factors such as the edge formation probability, the relation between the connected data samples, the noise level, or the importance to the downstream task.
>
> For example, as we discussed in Paragraph 3 of Section 1, in social networks, users are usually connected by some common characteristics, e.g. geological location, interests, or mutual friends. However, due to the random nature of social networks, there are also many connections from less relevant users, such as advertisers or even bots. These connections are less likely to contribute positively or even noisy to some prediction tasks such as political leaning prediction.
>
> In addition, in citation networks, two researchers with highly related research topics (e.g. machine learning and data mining) are more likely to collaborate with each other, while the reason behind a collaboration of two researchers with less related research topics (e.g. computer architecture and social science) might be more difficult to understand.
>
> Many other similar examples can be found in different domains such as brain network modeling [1], chemical property prediction [2], and biological network analysis [3].
>
>
> [1] Martijn P Van Den Heuvel and Hilleke E Hulshoff Pol. Exploring the brain network: a review on resting-state fmri functional connectivity. European neuropsychopharmacology, 20(8):519–534, 2010.
>
> [2] Saulo DS Reis, Yanqing Hu, Andrés Babino, José S Andrade Jr, Santiago Canals, Mariano Sigman, and Hernán A Makse. Avoiding catastrophic failure in correlated networks of networks. Nature Physics, 10(10):762–767, 2014.
>
> [3] Karplus, Martin, and Gregory A. Petsko. "Molecular dynamics simulations in biology." Nature 347.6294 (1990): 631-639.
>
> Review #2. i.d.d data is not correctly used. What the authors want to express through i.d.d data should be `independent data', in which the data have no connections. While i.d.d. data refers to different samples drawn independently from identical distributions.
>
> Response #2. We sincerely thank the reviewer for pointing out a more appropriate use of terminology. We have thoroughly revised the paper to address this issue by substituting `i.i.d' with the more appropriate terminology.
>
> Also, we would like to clarify that this terminology substitution would not affect the validity and robustness of our proposed curriculum learning strategy because our algorithm does not require any i.i.d assumption. The purpose of using the terminology was only to show the contribution of our work compared to previous works, where previous curriculum learning methods can only handle independent data while our method can handle dependent data with connections.
>
> Review #3. In experiments, it is unclear why the difficulty of the edges can correspond to the formation probability.
>
> Response #3. In the synthetic datasets, we specially designed the formation probability of an edge to reflect its likelihood to positively contribute to the node classification job, which indicates its difficulty level. Specifically, as we discussed in Section 5.1 and Appendix A.1, nodes with close classes are more likely to be connected since the formation probability decreases with the distance of the node label, and connections from nodes with close classes can increase the likelihood of accurately classifying a node due to the homophily property of the designed node classification task. Therefore, an edge with a high formation probability indicates a higher chance to positively contribute to the node classification task because it connects a node with a close class, and thus can be considered an easy edge.

---

> > ### Author Response · Authors · 2022-11-11
> > **Response to Reviewer yJ3m (2/4)**
> >
> > Review #4. The datasets are rather small.
> >
> > Response #4. Thank you for your suggestion, we have added additional experiments to evaluate the effectiveness of our proposed method on two large datasets. Also, we would like to clarify that the size of the datasets that we originally used in this paper is comparable to both our comparison methods papers [4-7] and many other publications [8-12] with related topics in recent years' top-tier machine learning conferences.
> >
> > For the added additional experiments, we choose ogbn-proteins and ogbn-arxiv datasets from the Open Graph Benchmark [13], where ogbn-proteins has 132,534 nodes and 39,561,252 edges, and ogbn-arxiv has 169,343 nodes and 1,166,243 edges. We follow the original train/validation/test split settings and report the accuracy/ROCAUC score on ogbn-arxiv and ogbn-proteins respectively according to the Open Graph Benchmark [13]. All models follow the same architecture with three convolution layers and hidden dimensions as 256. All classification results are averaged over 10 times running. The experimental results are shown below:
> >
> > -------------------------------------------------------------------------------------------------------------------------
> >             | GCN       | GNNSVD+GCN | CLNode + GCN| Ours + GCN
> >
> > ogbn-arxiv     | 71.74$\pm$0.29         | 67.44$\pm$0.51              | 70.95$\pm$0.18                | 74.08$\pm$0.33
> >
> > ogbn-proteins | 72.51$\pm$0.35 | 66.92$\pm$0.64       | 71.40$\pm$0.32         | 75.19$\pm$0.26
> >
> >             | GIN       | GNNSVD+GIN | CLNode + GIN| Ours + GIN
> >
> > ogbn-arxiv    | 69.26$\pm$0.34 | 67.79$\pm$0.41       | 70.58$\pm$0.17         | 72.55$\pm$0.15
> >
> > ogbn-proteins | 74.51$\pm$0.32 | 70.65$\pm$0.53       | 73.97$\pm$0.31         | 78.76$\pm$0.22
> >
> >            | GraphSage  | GNNSVD+GraphSage| CLNode + GraphSage| Ours + GraphSage
> >
> > ogbn-arxiv    | 71.49$\pm$0.27 | 69.82$\pm$0.34       | 71.54$\pm$0.20         | 73.13$\pm$0.14
> >
> > ogbn-proteins | 77.68$\pm$0.20 | 71.82$\pm$0.39       | 78.40$\pm$0.41         | 79.89$\pm$0.35
> >
> > -------------------------------------------------------------------------------------------------------------------------
> >
> >
> >
> >
> > We can observe that our method consistently outperforms all baseline models on two large datasets over three backbone GNN models. Specifically, our proposed method outperforms the second-best-performing model by 2.21\% on average. Besides, we have in 5 out of 6 cases the statistical significance $p<0.001$, and we have $p<0.01$ for the left case. It is also worth noting that other comparison methods used in our paper such as ProGNN, NeuralSparse, and PTDNet can not handle these two large datasets and went into an out-of-memory error.
> >
> > [4] Luo, Dongsheng, et al. "Learning to drop: Robust graph neural network via topological denoising." Proceedings of the 14th ACM international conference on web search and data mining. 2021.
> >
> > [5] Jin, Wei, et al. "Graph structure learning for robust graph neural networks." Proceedings of the 26th ACM SIGKDD international conference on knowledge discovery \& data mining. 2020.
> >
> > [6] Zheng, Cheng, et al. "Robust graph representation learning via neural sparsification." International Conference on Machine Learning. PMLR, 2020.
> >
> > [7] Chen, Yu, Lingfei Wu, and Mohammed Zaki. "Iterative deep graph learning for graph neural networks: Better and robust node embeddings." Advances in neural information processing systems 33 (2020): 19314-19326.
> >
> > [8] Liu, Xiaorui, et al. "Elastic graph neural networks." International Conference on Machine Learning. PMLR, 2021.
> >
> > [9] Mujkanovic, Felix, et al. "Are Defenses for Graph Neural Networks Robust?." Advances in Neural Information Processing Systems 35 (NeurIPS 2022) (2022).
> >
> > [10] Kim, Dongkwan, and Alice Haeyun Oh. "How to Find Your Friendly Neighborhood: Graph Attention Design with Self-Supervision." The Ninth International Conference on Learning Representations (ICLR 2021). International Conference on Learning Representations (ICLR 2021), 2021.
> >
> > [11] Chang, Heng, et al. "Not all low-pass filters are robust in graph convolutional networks." Advances in Neural Information Processing Systems 34 (2021): 25058-25071.
> >
> > [12] Liu, Xiaorui, et al. "Graph neural networks with adaptive residual." Advances in Neural Information Processing Systems 34 (2021): 9720-9733.
> >
> > [13] Hu, Weihua, et al. "Open graph benchmark: Datasets for machine learning on graphs." Advances in neural information processing systems 33 (2020): 22118-22133.

---

> > > ### Author Response · Authors · 2022-11-11
> > > **Response to Reviewer yJ3m (3/4)**
> > >
> > > Review #5. The proposed method is not consistently better than the baselines in all datasets. Besides, is there any experiment that can demonstrate that it is indeed the curriculum learning part that improves the performance, instead of other factors like model complexity?
> > >
> > > Response #5. In response to ''The proposed method is not consistently better than the baselines in all datasets.''
> > >
> > > 1. First, for the experiments on 7 original real-world datasets, as shown in Table 2, our proposed method achieved the best performance compared to all baseline models in 20 out of 21 tasks. For the only case where our method did not achieve the best performance, our method achieved the second-best performance, with only a 0.52\% gap in accuracy compared to the best-performing model, and the difference was weakly statistically significant with $p > 0.05$. Besides, the strong performance of our proposed method on the two additional large real-world datasets can further demonstrate that our model can consistently improve performance in real-world scenarios.
> > >
> > > 2. Second, for the experiments on synthetic datasets, as shown in Table 1, our proposed method achieved the best performance compared to all baseline models in 21 out of 27 prediction tasks. More importantly, in 5 out of the 6 cases where our method did not achieve the best performance, the homophily coefficient $\geq 0.7$ that almost all models can achieve near-perfect performance with classification accuracy close to 100\%, and the performance of our method is comparable to the best performance with only 0.11\% difference on average. Besides, for the only case that our method did not achieve the best performance when homophily coefficient $< 0.7$, our method achieved the second-best performance and largely outperformed the third-best method by 5.56\% in accuracy.
> > >
> > > 3. Third, we further added the statistical significance analysis on the experimental results in the revised paper, where in 37 out of 41 cases our method outperforms the second-best performing model with strong statistical significance. Specifically, we have in 25 out of 37 cases with a significance $p<0.001$, in 7 out of 37 cases with a significance $p<0.01$, and in 5 out of 37 cases with a significance $p<0.05$. Such statistical significance results can demonstrate that our proposed method can consistently perform better than the baseline models in all datasets.
> > >
> > > In response to ``Besides, is there any experiment that can demonstrate that it is indeed the curriculum learning part that improves the performance, instead of other factors like model complexity?''
> > >
> > > 1. First, we would like to clarify that our proposed method does NOT introduce any extra model parameters in prediction tasks. The model complexity of our model is exactly the same as the baseline backbone model. For example, if we use a two-layer GCN as the baseline model, then our proposed method would use the same model architecture of a two-layer GCN model (e.g. hidden dimension, dropout rate, etc) for a fair comparison.
> > > 2. Second, the search space for hyperparameters of our proposed method and all baseline models are the same for the purpose of a fair comparison.
> > > 3. Third, we have provided an ablation study in Appendix A.2 due to the limitation of space. In the ablation study, we removed the smoothing components mentioned in Section 4.3 from the proposed model. We can observe from the results that although the proposed smoothing techniques can further improve the model performance, the core part of our curriculum learning algorithm can still outperform the baseline models on all datasets by 1.93\% on average. The experimental results can prove that our curriculum learning strategy can improve the generalization ability of models.
> > > 4. Fourth, in Figure 3 of Section 5.4 we have provided visualization that further proved our learned curriculum strategy can effectively select the edges according to their difficulty level during the training process. Since the underlying difficulty level of edges in the synthetic datasets can be indicated by their formation probability, we classify the edges into three groups with balanced numbers according to their difficulties: easy, medium, and hard. From the visualizations of Figure 3, we can observe that our proposed curriculum learning strategy can effectively include most of the easy edges at the starting phase of training, and leave most of the hard edges until the end phase of training. Such selection behavior is highly consistent with the core idea of our method: training the model with easy edges first and then gradually including more hard edges into training.

---

> > > > ### Author Response · Authors · 2022-11-11
> > > > **Response to Reviewer yJ3m (4/4)**
> > > >
> > > > Review #6. The quality is questionable since some key points are not clear.
> > > >
> > > > Response #6. We sincerely thank the reviewer for pointing out the unclear parts and hope our response and the revised paper can address your concerns.
> > > >
> > > > Review #7. Reproducibility is unclear, and can only be checked by running the code.
> > > >
> > > > Response #7.
> > > > 1. We have shared the source code for the purpose of replication.
> > > >
> > > > 2. All the datasets we used in this paper can be publicly obtained from the Pytorch-geometric library [14].
> > > >
> > > > 3. We have included a paragraph of implementation details of our model in Appendix A.1 due to the space limit, where detailed requirements such as train/test split setting and computational resources can be found.
> > > >
> > > > 4. The searching space for hyperparameters of both our method and comparison methods can also be found in Appendix A.1 due to the space limit. We applied the same search space for our method and the baseline models for a fair comparison.
> > > >
> > > > [14] Fey, Matthias, and Jan Eric Lenssen. "Fast graph representation learning with PyTorch Geometric." arXiv preprint arXiv:1903.02428 (2019).

---

> > > > > ### Author Response · Authors · 2022-12-06
> > > > > **A warm reminder**
> > > > >
> > > > > Dear Reviewer,
> > > > >
> > > > > Thank you again for your time.
> > > > >
> > > > > The end date of Discussion Stage 2 is approaching, we hope that all your concerns have been addressed. If any concerns remain, we are happy to further clarify them. Looking forward to your reply.
> > > > >
> > > > > Best regards,
> > > > > Authors

---

### Decision · Program_Chairs · 2023-01-20

**Decision:**

Reject

**Justification For Why Not Higher Score:**

See the above-mentioned weaknesses.

**Justification For Why Not Lower Score:**

N/A

**Metareview: Summary, Strengths And Weaknesses:**

This paper introduces a curriculum learning strategy for GNNs by explicitly modeling the problem that edges in a graph may have different levels of difficulty to learn. It introduces a curriculum learning-based model to gradually incorporate edges during learning according to their difficulties. The difficulty level is obtained by self-supervised learning.

**Strengths:**
* Experiments are conducted on 9 synthetic datasets (with 9 different homophily coefficients) and 7 real-world datasets. However, the datasets are rather small.
* The paper is overall easy to understand. The motivation and the background knowledge of the curriculum learning are clearly introduced.

**Weaknesses:**
* Reviewers were concerned that the datasets are rather small. In response to this comment, the authors added additional experiments to evaluate the effectiveness on two larger datasets (ogbn-proteins and ogbn-arxiv).
* Reviewers also noted that it would be interesting to see how the model behaves when the graph homophily assumption breaks. In the rebuttal phase, the authors successfully clarified the method does not require a homophily assumption. It is a general learning framework, and any non-parametric kernels can be utilized in determining how well an edge is expected by its connecting nodes embeddings.
* Many ideas in the paper are not new. The overall loss function has three parts, where the first part is a supervised loss, the second part is a graph reconstruction loss, and the third one is a regularization term.  Although the authors introduce their method from a perspective of curriculum learning, Reviewers agree the contribution is limited as underlying ideas are similar to existing methods.
* The improvements over existing methods are not significant. The method does not perform consistently better than baselines on real-world datasets; the results on different datasets all have a high standard deviation.

The reviewers acknowledge the authors' rebuttal, including new experimental results added during the rebuttal. The rebuttal addressed some of the key concerns on the model performance. However, this paper needs further improvement. I suggest the authors address criticisms raised by the reviewers and refine the manuscript for a more solid publication.